# The Effect of a Nurse-Led Family Involvement Program on Anxiety and Depression in Patients with Advanced-Stage Hepatocellular Carcinoma

**DOI:** 10.3390/healthcare11040460

**Published:** 2023-02-05

**Authors:** Sukhuma Klankaew, Suthisa Temthup, Kittikorn Nilmanat, Margaret I. Fitch

**Affiliations:** 1Songklanagarind Hospital, Faculty of Medicine, Prince of Songkla University, Songkhla 90110, Thailand; 2Faculty of Nursing, Prince of Songkla University, Songkhla 90110, Thailand; 3Bloomberg Faculty of Nursing, University of Toronto, Toronto, ON M4C 4V9, Canada

**Keywords:** anxiety, depression, family involvement, hepatocellular carcinoma, palliative care

## Abstract

Psychological distress is commonly reported in patients with advanced cancer. Family is considered a psychological supporter for patients during their cancer journey. This study aimed to examine the effect of a nurse-led family involvement program on anxiety and depression in patients with advanced hepatocellular cancer. This is a quasi-experimental study with a two-group, pre–post-test design. Forty-eight participants were recruited at a male medical ward in a university hospital in Southern Thailand, and assigned to either the experimental or the control group. The experimental group received the nurse-led family involvement program, while the control group received only conventional care. Instruments included a demographic data form, clinical data form, and the Hospital Anxiety and Depression Scale. Data analyses were performed using descriptive statistics, chi-square, Fisher’s exact test, and *t*-test. The results revealed that the mean scores of anxiety and depression in the experimental group at post-test were significantly lower than on the pretest and significantly lower than those of the control group. The results indicate that a nurse-led family involvement program has a short-term effect on the reduction of anxiety and depression in male patients with advanced HCC. The program can be useful for nurses to encourage family caregivers to engage in patient care during hospitalization.

## 1. Introduction

Hepatocellular cancer (HCC) is a common cancer, in which most patients have a poor prognosis [1]. The disease is typically discovered late in the course of the illness, as it is challenging to identify in the early stages [2]. It was reported that between 15–20% of patients with HCC arrive at the doctor’s office when they are in an advanced disease stage and are apt to have a survival time of only 3–4 months after diagnosis [3]. Of these patients with advanced HCC, 94% die while in hospital [4]. Due to poor prognosis, major hepatology societies recommend integrating early supportive and palliative care for patients with HCC [1].

Psychosocial consequences are commonly reported among persons with cancer, although individuals may experience depression and anxiety differently, as a result of personal, psychological, social, and environmental factors as well as the type of cancer and medical treatments [5]. Anxiety and depression are reflective conditions regarding adaptation to the illness and may indicate the patient is having difficulty coping effectively with stress. Patients with advanced HCC have significant psychological distress [4]. A recent systematic review reported that 40% of patients with metastatic HCC have anxiety and depression [6] with HCC progression associated with depression [7]. Several other factors are related to anxiety and depression in patients with HCC, including female gender, higher Charlson comorbidity index scores, and liver cirrhosis [6]. In addition, advanced Barcelona Clinic Liver Cancer (BCLC) stage and undergoing liver resection are significantly related to more severe physical and psychological symptoms in these patients [2]. A systematic review focusing on factors associated with heightened risk for depression in cancer patients found that patients without personal relationships were up to four times more likely to experience depression compared with patients who were in relationships [8].

Furthermore, a recent systematic scoping review reported in patients with advanced cancer that higher levels of distress, depression, and anxiety were linked to higher levels of unmet need demands in the major domains (e.g., physical, emotional, practical) and across the broad spectrum of cancer types [9]. Wang et al. [10] examined unmet care needs of advanced cancer patients. Unmet psychological and physical needs were identified as two areas of focus for patients with advanced cancer, while family and friends’ support was the most common unmet social need.

In Thailand, nearly 40% of patients with HCC who come to see a doctor have already entered an advanced stage of disease [11,12]. These patients are often hospitalized and receive palliative care to relieve distressful symptoms. However, being hospitalized may result in experiencing a sense of loneliness and social isolation. Riedl and Schüßler [8] reported that depression and emotional distress were consistently related to social deprivation and poor social support. Recently, Temtap and Nilmanat [13] found that hospitalized patients with advanced HCC experienced high levels of anxiety and depression.

In addressing the need for psychological care for cancer patients, Weis [14] suggested a stepped-care approach, including systematic need assessment, integrated psychosocial care delivery by care managers that range from counseling to individual psychotherapy, and appropriate professional supervision. A randomized control trial of a comprehensive education and care program was found to be beneficial in reducing anxiety and depression in patients with hepatocellular carcinoma who underwent surgery [15]. In addition, a recent systematic review reported beneficial effects of psycho-oncological intervention on the reduction of anxiety and depression symptoms for this population [6].

Family support is a common form of social support and has been reported to enhance effective coping strategies [16,17] and reduce anxiety among patients with cancer [18]. Family members may be in the best position to provide emotional support to these patients during their cancer journey. It is suggested that people living with cancer may benefit from psychosocial interventions that are focused on patient and family caregivers [19]. However, at present, family involvement in an acute care hospital setting is characterized by poor communication, a lack of agreement of caregiving roles, and a lack of cooperation and collaboration between family caregivers and nurses [20]. Culture also influences the level of family involvement in treatment decision making, especially for older adults with cancer [21].

To date, several psychological interventions have been conducted in patients with some cancer types, such as prostate cancer [22,23] and breast cancer [24,25]. However, there is a paucity of literature on family involvement interventions that focus on psychosocial aspects, particularly in adult acute care hospital settings. Thailand is a family-oriented society, and family caregivers are often present in healthcare facilities during a patient’s hospitalization. Frequently, family members are used for inpatient care. To our knowledge, no other intervention study has provided a family engagement program to reduce psychological distress among Thai patients with HCC. We anticipated that an intervention program to promote family involvement could fulfill the care need, consequently alleviating anxiety and depression in patients with advanced cancer.

### Conceptual Framework and Literature Review

This study used the family support concept [26] and the literature review on the needs of patients with HCC, as well as the roles of nurses in palliative care [17,27,28,29,30] to develop an intervention program, namely the nurse-led family involvement program. Furthermore, the family involvement program was designed and centered on facilitating communication within the family. Previous studies found that a communication-focused approach enhanced the quality of life and coping skills of family caregivers for patients with cancer [27].

People with advanced HCC experience physical symptoms as well as difficulty performing daily tasks. They, therefore, have a need for assistance. Hien, Chanruangvanich, and Thosingha [31] examined palliative care needs among patients with HCC. They found that patients who have high levels of physical symptoms, anxiety, and depression but low social support would have high palliative care needs [31]. Palliative care support for patients with HCC includes symptom management, decisional support, care coordination, and psychosocial support [1,32]. Nurses can provide palliative care to patients with HCC and their families at any stage of the disease.

In order to respond to the palliative care needs of patients with HCC holistically, we consider families as the main source of support for patients with HCC. Family support can be given in the form of information, emotional support, and practical assistance to sick persons [26]. Therefore, our program involved family members of patients with HCC to engage in four domains of care activities, including information sharing, care decisions, care provision, and psychological support. Nurses facilitate family involvement during visiting time. It was recommended that healthcare providers must acknowledge the presence of the family and routinely interact with them [28]. Information sharing is one of the education sessions. Nurses provide health education tailored to patients’ information needs. Evidence reveals that patients may feel difficulty comprehending disease-specific information provided by medical doctors [33]. Family members help patients in remembering or recalling information, giving information directly to the doctor, or clarifying instructions from the doctor so the patient can understand and seek necessary information [34]. The program facilitates patients and family caregivers to interact with the nurse and ask questions. Furthermore, a previous systematic study found that fatigue was reported as the most prominent physical unmet need among patients with advanced cancer [10]. Fatigue was a common physical health problem experienced by patients with HCC [35,36]. Nurses involve families to provide bedside care for patients with HCC.

Effective communication and shared decision making were reported as one of the most important elements in end-of-life care [29]. Similar to many Asian patients, Thai families played a crucial role in treatment decisions. These patients preferred to share decisions about cancer treatments with their family [30]. A previous study found that making decisions alone was linked to lower emotional, social, and spiritual wellbeing among Asian patients with advanced cancer, whereas making decisions jointly with doctors and family was linked to greater social and spiritual wellbeing [37]. Spending more time discussing treatment choices with family members helps patients cope with their cancer diagnosis and promotes cognitive processing, both of which may eventually reduce the patient’s stress levels [38]. Psychological support is another aspect of the program. Nurses encourage the family to provide emotional and psychological support to their loved ones. It was found previously that during cancer treatment, patients frequently did not want to express difficult feelings with nurses [39]. Patients may be more willing to share their concerns with their family members. Support, particularly emotional support, from family members is crucial for coping with cancer [17]. They believed that their families were crucial in helping them cope with their condition and face the doubts and anxieties that came with receiving a cancer diagnosis [40].

From a literature review, the family support and palliative care concept seems to be appropriate for guiding an intervention to reduce anxiety and depression in patients with advanced-stage HCC. Therefore, we were interested in examining the effect of a nurse-led family involvement program to reduce anxiety and depression of patients with advanced HCC.

## 2. Materials and Methods

### 2.1. Design

This study used a quasi-experimental study with a two-group, pre–post-test design. The purpose of this study was to compare anxiety and depression between the experimental group who received a nurse-led family involvement program and the control group who received only conventional care. This report follows Transparent Reporting of Evaluations with Non-randomized Designs (TREND) [41].

### 2.2. Participants

The authors recruited persons with HCC who met the inclusion criteria at a male medical unit in a university hospital in Southern Thailand. The inclusion criteria for persons with cancer were (1) knowing their diagnosis of advanced hepatocellular cancer according to the Diagnostic Statistical Manual-IV diagnostic criteria (BCLC stage C) or (BCLC stage D) and receiving palliative care treatments, (2) being fully conscious and able to communicate in Thai, and (3) having a primary family caregiver to provide care continuously in the hospital. The exclusion criteria were (1) being critically ill and experiencing severely acute exacerbations such as difficulty breathing or acute renal failure, (2) declining to participate in the study, and (3) having an anxiety or depression score higher than 11 on the Thai version of Hospital Anxiety and Depression Scale (HADs-Thai version) [42].

The sample size was calculated using Polit and Beck’s [43] recommendation that the minimum sample size for a quasi-experimental study was 20–30 subjects. In this study, the sample size was determined as 20 per group, with an additional 20% in each group to account for any dropout of subjects. A total of 48 eligible patients were recruited (24 subjects per group) by the researcher and allocated to the experimental or control groups using a simple random allocation method. During the study, 8 subjects were excluded and withdrawn from the study due to high psychological distress (3) and severity of disease (5), respectively, leaving 20 subjects per group.

For each patient, a family member was also recruited. The inclusion criteria for the family members were (1) being a primary caregiver who provides care continuously for the patient during the cancer treatment process in the hospital, (2) being aware of the advanced stage of cancer of the patient, (3) being 18 years and older, (4) being willing to participate in the study and able to attend all sessions of the four-day intervention program (for experimental group), and (5) being able to read and understand the Thai language.

### 2.3. Interventions

All patient participants received conventional nursing care according to the clinical practice and palliative care guidelines in our Palliative Care Manual of the Nursing Service Department. The conventional care included routine monitoring of vital signs and hemorrhage signs; routine blood and urine laboratory examinations; management of discomfort symptoms, diet, activity, personal hygiene care, and so on.

The nurse-led family involvement program was developed by the research team based on a literature review regarding the needs of patients with advanced cancer and the concept of family support and palliative care [17,26,27,28,29,30]. The program contents and activities were reviewed and validated by three experts and revised as recommended before implementation. The nurse-led family involvement program intervention focused on four aspects of family involvement, including information sharing, care decisions, care provision, and psychological support. Both the patient and family member received the intervention together on the first day of hospitalization and continued for four days in a row. Each session was held at the patient’s bedside and took between 30–60 min during family visiting time. A member of the research team provided the intervention. In addition, a caregiver booklet on caring for patients with advanced HCC was provided. The content of the booklet included the caregivers’ role in supportive caring activities, self-care activities, symptoms, and symptom management in HCC. The details of the program are presented in Table 1.

### 2.4. Instruments

The instruments utilized to gather data on demographic characteristics of patients and family caregivers as well as clinical data were developed by the PI for the purposes of the study. The items collected data on (1) patient characteristics, including age, religious affiliation, educational level, marital status, occupation, sufficiency of income to cover current expenses, and types of health insurance; (2) family caregivers’ characteristics, including gender, age, educational level, occupation, and relationship with patient; and (3) patient clinical data, including Thai Palliative Performance Scale score [44], the severity of the disease with Child Pugh score [45], and times since diagnosis.

The instrument used to collect anxiety and depression data was the Hospital Anxiety and Depression Scale (HADS), developed by Zigmond and Snaith [46]. This self-rating scale has two subscales: one measures depression with seven items and the other measures anxiety with seven items. Each item is scored on a 4-point (0–3) Likert scale and rates how the individual has been feeling in the past week. The scores of 0–7 indicate “normal”, while 8–10, 11–14, and 15–21, “indicate mild”, “moderate”, and “severe”, respectively. In this study, the Thai version of HADs was used [42], and the Cronbach’s alpha coefficient was found to be 0.89 for the anxiety subscale and 0.82 for the depression subscale.

### 2.5. Ethical Considerations

The study was conducted in accordance with the Declaration of Helsinki and was approved by the Institutional Review Board of the Faculty of Medicine, Prince of Songkla University (EC 62-131-15-7). All eligible subjects were informed of the objectives and processes of the study, the benefits and potential risks, time required, rights to privacy, confidentiality, and ability to withdraw without losing healthcare service benefits. They were given opportunities to ask questions. All subjects signed consent forms before the study participation. Furthermore, written informed consent has been obtained from the patients to publish this paper.

### 2.6. Data Collection and Intervention

After IRB approval, ward nurses informed the research team about new admission of patients with HCC. A member of the research team (ST) then recruited patients who met the inclusion criteria and agreed to participate in the study. She subsequently introduced herself to the patients’ family caregivers and invited them to participate in the study. Participants in both groups were informed about the study before signing the informed consent form.

After consent was obtained, patients and family caregivers were asked to complete the demographic data. The experimental group then received the family involvement program and conventional care. The PI (SK) provided the intervention to the experimental group. The control group received conventional care.

Two research assistants (RA) who were nurses from the internal medicine ward and trained for data collection and research ethics collected the data. The RA completed the clinical data form and HAD questionnaires in both patient groups at the pretest on the first day of hospitalization and post-test on the fifth day of hospitalization. RAs were unaware of the group assignment.

Of the 48 patients with advanced HCC assessed for eligibility, all were invited to participate in the study. However, only 23 patients in the experimental group and 22 patients in the control group met the inclusion criteria. During the study, three participants in the experimental group and two in the control group withdrew because of progression in severity of disease. A total of 40 participants completed the study, with 20 in the experimental group and 20 in the control group (Figure 1). Data were collected from July 2019 to August 2021.

### 2.7. Data Analysis

Data were analyzed using R program. Descriptive analysis was performed for all variables and presented as mean and standard deviation, count (percentage), or median and interquartile range. To compare differences between the two groups at baseline, chi-square statistical analyses and Fisher’s exact test were employed. Comparison between groups was determined by paired *t*-test, independent-sample *t*-test, Chi-square test, or Wilcoxon rank sum test depending on the type of data.

## 3. Results

Forty participants completed the study. Twenty participants were in both the experimental and control groups. Demographics regarding age, marital status, educational level, occupation, age of caregivers, relationship with caregiver did not differ statistically between experimental and control groups. For clinical data, PPS scores, Child Pugh scores, and time since diagnosis were not significantly different between the two groups (Table 2).

The means and standard deviations for anxiety and depression in both groups at baseline are presented in Table 3. The independent *t*-test was used to compare the scores of two outcome variable measures at baseline. There were no significant differences in the overall mean scores for anxiety and depression between experimental and control groups.

Following the intervention program, the experimental group’s mean scores for anxiety and depression decreased significantly at post-test compared to baseline (Table 4).

The post-test comparison between the experimental and control groups found that the mean scores for anxiety and depression in the experimental group were significantly lower than those of the control group (Table 5).

## 4. Discussion

This study was undertaken to evaluate the effectiveness of a program given to patients with advanced cancer and family caregivers for reducing anxiety and depression in patients. Hospitalized patients are apt to have high psychosocial distress, and family members are in a good position to influence that distress. Especially in Thailand, there is a large presence of family members in the hospital. Findings from this study demonstrated that the family involvement program reduced anxiety and depression in patients with advanced HCC. The program consisted of four aspects of family involvement, including health information sharing, care provision, decision making on treatments, and providing psychological support.

In this program, nurses were encouraged to build rapport and establish trusting relationships with patients and family during the first day of hospital admission. This is aimed at affirming an important role of families [28]. The nursing activities included encouraging patients and their caregivers to be open and share concerns as well as impacts of the cancer experiences on all family members. In the information sharing session, the nurse provided education and support tailored to patients’ needs and preferences for the level of required information. She also invited family caregivers to participate and provided the booklet for patients and other family members to learn. Family members could help patients manage physical symptoms and cope with psychological concerns. In addition, the nurse supported family members to engage in conversations with the physician. A previous study reported that health information assists patients in coping with the immediate and long-term physical, emotional, and social impacts of cancer [47].

Advance care planning is a key component of palliative care for patients with HCC [29]. Our nurse-led program facilitated family caregivers to become involved in decision making through providing and sharing information with the patient, coordinating care with medical doctors for consultation, and setting family meetings among the patient, family members, and healthcare providers. Patients often viewed their family members as supporters to help cope with cancer and treatment options and identify preferences [21] and mentioned that it was important to discuss or share any decisions made with their caregivers [48]. Therefore, these nursing activities would alleviate patients’ psychological distress.

Another aspect of our nurse-led program was to encourage family caregivers to be involved in care provision. Based on PPS assessment, most of our study participants were in the transition phase. For the most part, these patients were unable to perform activities of daily living. Encouraging family caregivers to be involved in daily patient care such as bathing, grooming, or eating can fulfill the physical needs of their loved ones. Furthermore, during care involvement, patients and family caregivers spend time together, which can enhance the intimate relationship and promote a sense of connectedness and alleviate feelings of loneliness [40].

Our findings confirmed the previous systematic review, which reported that psycho-oncological intervention can alleviate depressive symptoms and anxiety in patients with HCC [6] and patients with prostate cancer [22,23] and with breast cancer [24,25]. This nurse-led program is appropriate for an inpatient setting. In general, the length of stay of hospitalized patients with HCC in the selected hospital was five to six days. Therefore, our brief one-on-one family involvement program could be beneficial for both health care providers and patients as well as their family. It can be provided during hospital visiting times each day. It has been reported that a brief intervention, delivered by lay persons, can promote adjustment among newly diagnosed cancer patients at high risk of developing anxiety or depressive disorders [49]. Nurses take roles to facilitate interaction and communication among patients, family, and health care providers. Based on our observations during implementation of the program, we witnessed happy moments and saw the sparking eyes among patients. Instead of lying on their beds all day, these patients looked energized. Our study participants also reflected that they spent their time together meaningfully. When all aspects of the intervention are provided in a synchronized manner, they can alleviate psychological distress as experienced by these patients.

## 5. Limitation of the Study

There are some limitations in this study. The participants were family caregivers of male patients with advanced HCC in the university hospital, Southern, Thailand. Therefore, generalization to other settings and female patients is limited. This study was conducted during the COVID-19 pandemic, hence there was a limited number of eligible participants in the study. It would be valuable to repeat the study with a larger sample as well as with a female patient sample. Finally, the outcomes were assessed immediately after the program completion. Interventions cannot be assumed to have long-term effects. Moreover, only patient outcomes were measured. Further studies should be undertaken to examine the outcomes on family caregivers and measure the outcomes over a more extended period in both patients and family caregivers. Due to the complexity of care demands among other hospitalized patients, implementing the program in other inpatient settings may require adjusting the intervention to make it more practical in real-world settings.

## 6. Conclusions and Implications

Depression and anxiety are common among patients with HCC. The study addressed the significance of family involvement in alleviating anxiety and depression in hospitalized patients with advanced HCC. The results show that our nurse-led family involvement program had a short-term effect of reducing anxiety and depression symptoms immediately after intervention. This program was delivered successfully by registered nurses to patients with moderate levels of psychosocial distress. In the future, to implement the program in medical settings, nurses will require preparation for screening psychological distress and providing the intervention through orientation to the program activities.

## Figures and Tables

**Figure 1 healthcare-11-00460-f001:**
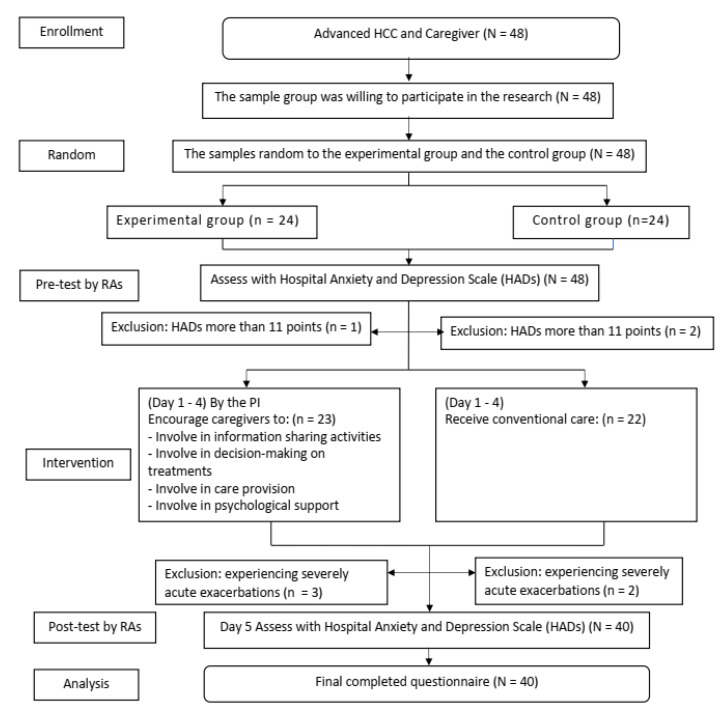
Flow diagram of the participants in the study.

**Table 1 healthcare-11-00460-t001:** Schedule and contents for the 4-day nurse-led family involvement program.

Day	Duration (Minutes)	Key Aspects of Involvement	Activities
1	15–20	Pretest	- Collecting HADs questionnaire from patients by RAs
	30–40	- Information sharing - Care provision - Psychological care	- Establishing relationship and building trust with the patient and their family caregiver - Encouraging them to share experiences and to express feelings and thoughts related to illness and care needs - Addressing the significant roles of family caregivers along the cancer journey - Providing information about the disease, treatment plans, and care demands of patients to the patient and their family caregivers - Providing family caregivers a booklet on caring for patients with advance HCC - Encouraging family caregivers to engage in care provision of the patients, including personal hygiene care, comfort care
2–4	40–60	- Information sharing - Care decisions - Care provision - Psychological care	- Involving family caregivers during providing health information to the patients and encouraging them to ask questions if there are any - Explaining about patients’ symptoms, treatment, and care plan, and encouraging family caregivers to review about patient’ s problems - Involving family caregivers in conversations with the physicians - Facilitating open discussion among family and listening attentively - Consulting the family physician to initiate a family meeting and share any decisions - Encouraging family caregivers to be involved in care provision for the patients such as bed baths, comfort care - Providing information related to psychological care activities to family caregivers, such as listening, use of touch, being present, using positive words, or relaxation techniques
5	15–20	Post-test	- Collecting HADs questionnaire from patients by Ras

**Table 2 healthcare-11-00460-t002:** Basic characteristics of patients with HCC in the experimental and control groups (N = 40).

Characteristics	Group		Statistic Results	*p*-Value
	Intervention (20) *n*(%)	Control (20) *n*(%)		
Religious affiliation ^#^			0.404	0.525
Buddhism	10 (50)	12 (60)		
Islam	10 (50)	8 (40)		
Educational level ^#^	6 (30) 8 (40) 6 (30)	5 (25) 9 (45) 6 (30)	0.150	0.928
Primary school				
Secondary school/high school/diploma				
Bachelor degree/or higher				
Marital status ^##^			1.619	0.695
Single	1 (5)	1 (5)		
Married	17 (85)	14 (70)		
Widowed/divorced/separated	2 (10)	5 (25)		
Occupation ^##^			0.648	0.878
Unemployed	15 (75)	13 (65)		
Government officer	3 (15)	5 (25)		
Merchant	2 (10)	2 (10)		
Caregivers’ age (years) ^#^			0.784	0.661
<60	16 (80)	18 (90)		
≥60	4 (20)	2 (10)		
Relationship with patients ^#^			0.114	0.736
Spouse	14 (70)	13 (65)		
Daughter/son	6 (30)	7 (35)		
Caregivers’ gender ^##^			0.173	0.500
Female	16 (80)	17 (85)		
Male	4 (20)	3 (15)		
Caregivers’ occupation ^#^			3.552	0.314
Farmer/gardener	9 (45)	12 (60)		
Merchant	9 (45)	4 (20)		
Government officer	2 (10)	3 (15)		
Unemployed	0	1 (5)		
PPS ^##^			1.726	1.000
End-of-life phase	0 (0)	1 (5)		
Transition phase	18 (90)	18 (90)		
Stable phase	2 (10)	1 (5)		
Severity of disease (Child Pugh			0.100	0.752
score) ^#^				
Child Pugh score B	11 (55)	10 (50)		
Child Pugh score C	9 (45)	10 (50)		
Duration since diagnosis with			1.245	0.776
advanced stage (days) ^##^				
≤30	4 (20)	6 (30)		
31–60	8 (40)	5 (25)		
61–90	5 (25)	5 (25)		
>90	3 (15)	4 (20)		

^#^ Pearson Chi-square test ^##^ Fisher’s exact test.

**Table 3 healthcare-11-00460-t003:** Comparison of anxiety and depression scores at baseline between experimental and control groups (N = 40).

Variables	Group	Mean	SD	df	t	*p*-Value
Anxiety	Experiment	10.65	0.67	19	0.24	0.813
	Control	10.70	0.65			
Depression	Experiment	10.40	0.82	19	1.59	0.119
	Control	9.70	1.78			

**Table 4 healthcare-11-00460-t004:** Comparison of anxiety and depression scores at each point of measurement in experimental group (*n* = 20).

Variables	Time Point	Mean	SD	df	t	*p*-Value
Anxiety						
	Pretest	10.65	0.67	19	7.55	0.000 *
	Post-test	9.15	0.81			
Depression						
	Pretest	10.40	0.82	19	5.44	0.000 *
	Post-test	8.80	1.10			

* *p* < 0.001.

**Table 5 healthcare-11-00460-t005:** Comparisons of anxiety and depression scores after intervention between experimental and control groups (N = 40).

Variables	Group	Mean	SD	df	t	*p*-Value
Anxiety	Experiment	9.15	0.81	38	4.05	0.000 *
	Control	10.15	0.74			
Depression	Experiment	8.80	1.10	38	2.59	0.013 **
	Control	9.80	1.32			

* *p* < 0.0001, ** *p* < 0.005.

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
