# Peer review of "The Effect of a Nurse-Led Family Involvement Program on Anxiety and Depression in Patients with Advanced-Stage Hepatocellular Carcinoma"

_healthcare, 2023, doi:10.3390/healthcare11040460_

Round 1

Reviewer 1 Report

Although relevant, the presented study is focused on Thailand and the characteristics of the Thai family. It would be important to present a comparative approach with what happens in the treatment of this type of cancer in other countries and the type of approach that is taken in these different countries.

The Introduction is adequate, but there is no section dedicated to reviewing the literature, which would be important to deepen the discussion on the subject, especially regarding the role of nurses in terms of their involvement in the recovery of patients and in the monitoring of families. Conclusions and implications are insufficiently developed.

Author Response

Thank you for your comments. We have revised the work point by point, as attached.

Reviewer 2 Report

In the introduction a deeper focus on simultaneous palliative care realted to patients' population considered could be useful to strenghten the study rationale

The part 2.1 Design could be improved with more references in bibliography about the method

In the discussion the major topics are frequently repeated; this part could be narrowed, together with more bibliography to support discussion

Author Response

(The authors gave the same response as above.)

Reviewer 3 Report

Manuscript is fairly detailed, but Authors missed inconsistent data between results and discussion. I suggest a better integration betweend intruduction, aim, findings, discussion.

So, based on yours findings, how do we proceed with managing anxiety and depression in patients with advanced Hepatocellular Cancer? Do you feel the current data (yours and others) support this programm? Please elaborate and discuss.

Author Response

(The authors gave the same response as above.)
